# Implicit Subgraph Neural Network

**Yongjian Zhong** [1]   **Liao Zhu** [1]   **Hieu Vu** [1]   **Bijaya Adhikari** [1]

## Abstract

Subgraph neural networks have recently gained prominence for various subgraph-level predictive tasks. However, existing methods either *1)* apply simple standard pooling over graph convolutional networks, failing to capture essential subgraph properties, or *2)* rely on rigid subgraph definitions, leading to suboptimal performance. Moreover, these approaches fail to model long-range dependencies both between and within subgraphs—a critical limitation, as many real-world networks contain subgraphs of varying sizes and connectivity patterns. In this paper, we propose a novel implicit subgraph neural network, the first of its kind, designed to capture dependencies across subgraphs. Our approach also integrates label-aware subgraph-level information. We formulate implicit subgraph learning as a bilevel optimization problem and develop a provably convergent algorithm that requires fewer gradient estimations than standard bilevel optimization methods. We evaluate our approach on real-world networks against state-of-the-art baselines, demonstrating its effectiveness and superiority. Our code is avaliable https://github.com/MLonGraph/ISNN

## 1. Introduction

Graphs serve as natural abstractions for complex relational data across numerous domains, including but not limited to the web (Kumar et al., 2000; Kleinberg et al., 1999), healthcare (Wang et al., 2020), social networks (Leskovec et al., 2007; Ugander et al., 2011), language and speech (Nastase et al., 2015), and e-commerce (Huang et al., 2004). A subgraph of a graph $\mathcal{G}(\mathcal{V}, \mathcal{E})$ is another graph $\mathcal{S}(\mathcal{V}', \mathcal{E}')$ induced by a subset $\mathcal{V}' \subset \mathcal{V}$ of nodes, where naturally $\mathcal{E}' \subset \mathcal{E}$.

Subgraphs capture critical internal structures within graphs, which are essential for a variety of applications. For example, echo chambers in social networks are subgraphs vital for studying misinformation diffusion and polarization (Cinelli et al., 2021); fraudulent activities within e-commerce networks induce subgraphs with properties that distinguish them from the rest of the graph (Song et al., 2021); and cascades of hospital-acquired infections in healthcare networks create subgraphs that aid in detecting asymptomatic infections (Jang et al., 2021).

The ubiquity of graphs and relational data has driven advancements in machine learning algorithms tailored to graphs, with graph neural networks (Wu et al., 2020) emerging as the dominant method across many, if not all, of these domains. While most current graph neural networks focus on node, edge, and graph-level tasks, there have been some, albeit limited, advancements in subgraph neural networks and subgraph representation learning.

(Adhikari et al., 2018) proposed `Sub2Vec`, a subgraph-level representation learning framework that captures a subgraph's neighborhood and structural information in a latent space. However, the approach is simplistic and fails to achieve state-of-the-art performance. (Alsentzer et al., 2020) introduced Subgraph Neural Networks (`SubGNN`), one of the first neural networks explicitly designed for subgraphs, incorporating positional information (i.e., where a subgraph is located within the entire graph) alongside neighborhood and structural properties. However, `SubGNN` relies on artificially sampled patches and, as later studies show, offers only marginal improvements over standard GNNs.

Subsequent works, `GLASS` (Wang & Zhang, 2021) and `SSNP` (Jacob et al., 2023), focus on pooling node-level representations to learn subgraph embeddings. `GLASS` employs a "labeling trick" to indicate node membership in a subgraph, learning node embeddings that are then aggregated into subgraph representations. `SSNP` first transforms node embeddings and applies multi-step stochastic pooling to generate subgraph embeddings. Notably, both methods rely solely on node memberships and overlook additional subgraph-level information.

Moreover, capturing long-range dependencies between subgraphs is critical in subgraph representation learning, as subgraphs that are far apart in the graph may still share

[1]Department of Computer Science, University of Iowa, Iowa City, USA. Correspondence to: Yongjian Zhong <yongjian-zhong@uiowa.edu>, Bijaya Adhikari <bijaya-adhikari@uiowa.edu>.

*Proceedings of the 42nd International Conference on Machine Learning*, Vancouver, Canada. PMLR 267, 2025. Copyright 2025 by the author(s).

complementary structural information that enhances classification performance. None of the prior work explicitly captures long-range dependency.

To address these issues, we first propose incorporating label-aware subgraph-level information alongside node embeddings to learn more expressive subgraph representations. Leveraging subgraph-subgraph relations provides an additional advantage (see Figure 3). As shown in the figure, explicitly modeling these relationships reveals hidden connections between subgraphs that may not be apparent in the original graph. While incorporating subgraph-subgraph relations allows our model to capture certain long-range dependencies, it remains insufficient for ensuring general long-range interactions across subgraphs. To fully address this, we formulate our subgraph neural network using the Implicit Neural Network framework (Sitzmann et al., 2020).

Implicit graph neural networks (IGNNs) have gained popularity for capturing long-range dependencies while avoiding performance degradation (Baker et al., 2023; Liu et al., 2022). At a high level, IGNNs iterate a single graph convolution operator until the learned node representations converge to a fixed point. Since the number of iterations is unbounded, the representations can incorporate information from topologically distant nodes. While various IGNNs exist—including models for dynamic (Zhong et al., 2024) and heterogeneous graphs (Gu et al., 2020)—there are currently no implicit models designed for subgraphs. Moreover, conventional IGNNs cannot be trivially adapted to subgraph learning, as they become unstable when approaching the fixed point (see Section 4.2).

**Present work.** In this paper, we propose the IMPLICIT SUBGRAPH NEURAL NETWORK (ISNN), the first implicit model for subgraph representation learning (see Figure 2). Our approach consists of an IGNN applied to both the original graph and a newly constructed hybrid graph. This hybrid graph extends the original structure by incorporating components that explicitly model relationships between subgraphs, as well as interactions between subgraphs and original graph nodes.

However, training such a model presents significant challenges (see Section 4). To address this, we reformulate the problem as a bilevel optimization problem (Sinha et al., 2017), where the upper-level objective corresponds to the subgraph classification task, while the lower-level constraints enforce the fixed-point conditions of the implicit representation. Although optimizing non-convex functions in bilevel settings is generally difficult, our formulation includes non-expansive constraints, enabling us to develop an efficient bilevel optimization algorithm. This algorithm leverages fixed-point iteration instead of gradient descent in the inner loop, reducing gradient oracle calls. We further provide a theoretical convergence guarantee, ensuring that

the algorithm reaches a stationary solution when one exists.

Our key contributions are as follows:

- We introduce **the first implicit subgraph neural network**, leveraging both node-level and label-aware subgraph-level information.

- We propose an **efficient bilevel optimization algorithm** for training ISNN, replacing gradient descent in the inner loop with fixed-point iteration, reducing gradient complexity while ensuring convergence.

- We conduct extensive experiments on **four benchmark datasets**, demonstrating that our approach consistently outperforms state-of-the-art baselines in F1-score across all datasets and in AUROC across all but one dataset.

## 2. Related Work

**Subgraph Neural Networks:** Initial works like Sub2Vec (Adhikari et al., 2018) introduced subgraph-level representation learning by focusing on capturing two important properties of subgraphs - structural and neighborhood. However, its simple architecture limited its expressiveness, leading to suboptimal performance compared to more sophisticated graph neural networks. Subgraph Neural Networks (Sub-GNN) (Alsentzer et al., 2020), was one of the first attempts to explicitly account for positional information of subgraphs within the entire graph, as well as their local structural features. To capture the full topologies of subgraphs, SubGNN samples anchor patches from the base graph and propagates messages between the anchors and the subgraph across multiple channels. However, this process of sampling channel-specific anchor patches and then propagating them is computationally intensive.

Recent approaches, such as GLASS (Wang & Zhang, 2021) and Stochastic Subgraph Neighborhood Pooling (SSNP) (Jacob et al., 2023), have focused on improving subgraph representation learning by using node-level representations. These methods first employ node embeddings then pool them to form subgraph embeddings. GLASS utilizes a novel labeling trick to distinguish whether a node belongs to a subgraph, and SSNP offers a multi-step stochastic pooling method to process node embeddings. However, even with these advances, both models focus mainly on node memberships and do not fully utilize other subgraph-level properties, such as structural or relational information between subgraphs.

**Graph Implicit Models:** Implicit models define their output through fixed-point equations. (Bai et al., 2019) provides an equilibrium model for sequence data based on an equilibrium equation's fixed-point solution. Implicit Graph Neural

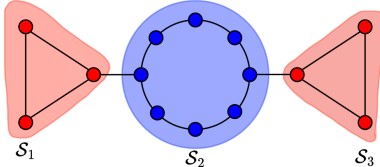

(a) A graph with three subgraphs

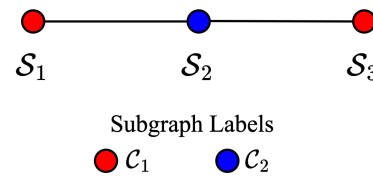

(b) Corresponding subgraph-level graph

Figure 1: The distance between the two subgraphs - $S_1$ and $S_3$ - in the original graph (a) is approximately $\frac{n}{2}$ (where $n$ is the number of nodes in the blue circle). However, in the subgraph-level graph (b) the distance for the same two subgraphs is only 2.

Network (IGNN) (Gu et al., 2020) showcases an implicit model, leveraging several layers of graph convolution network (GCN) to discover long-range dependencies. (Park et al., 2021) introduce the equilibrium GNN-based model with a linear transition map while ensuring the transition map is contracting such that the fixed point exists and is unique. (Liu et al., 2021a) present an infinite-depth GNN which captures long range dependencies in the graph while avoiding iterative solvers through deriving a closed-form solution. (El Ghaoui et al., 2021) present a general implicit deep learning framework and discusses the well-posedness of implicit models. (Chen et al., 2022) employ the diffusion equation as the equilibrium equation and solve a convex optimization problem to find the fixed point in their model.

**Bilevel Optimization:** Bilevel optimization has gained significant popularity in recent years due to its applicability in various domains. A bilevel problem involves two nested optimization tasks: an upper-level problem and a lower-level problem, where the upper-level decision depends on the solution of the lower-level problem. These problems are generally difficult to solve. For bilevel problems with a strongly-convex lower-level problem, algorithms based on the Implicit Gradient (IG) method have been proposed, as seen in the works of (Pedregosa, 2016; Ghadimi & Wang, 2018). Another branch of bilevel optimization is Iterative Differentiation (ITD) methods, which approximate the lower-level solution using iterative optimization techniques, as discussed by (Liu et al., 2021c; Hu et al., 2022). Additionally, penalty-based methods, such as log-barrier and gradient norm penalty, have been introduced to solve bilevel problems, as explored by (Liu et al., 2021b; Shen & Chen, 2023). Bilevel optimization has also been successfully applied in various fields, including federated learning (Tarzanagh et al., 2022) and ranking (Qiu et al., 2022).

## 3. Methodology

### 3.1. Preliminaries

**Notation:** Consider a graph $G = \{\mathcal{V}, \mathcal{E}, \mathbf{X}\}$, where $\mathcal{V}$ represents the set of nodes, $\mathcal{E}$ represents the set of edges, and $\mathbf{X} \in \mathbb{R}^{n \times d}$ is the feature matrix. Additionally, let $n$ be the number of nodes and $d$ be the feature dimension. We are given a set $\mathcal{D}$ of subgraphs drawn from $G$ and their corresponding labels, i.e., $\mathcal{D} = \{(\mathcal{S}_i(\mathcal{V}_i, \mathcal{E}_i), y_i)\}_{i=1}^m$, where $\mathcal{V}_i \subseteq \mathcal{V}$ is a subset of nodes and $\mathcal{E}_i$ is the edges induced by $V$ (naturally, $\mathcal{E}_i \subset \mathcal{E}$), and $m$ is the total number of subgraphs. This work focuses on the task of subgraph classification, where the goal is to predict the correct label $y_i$ for each subgraph $\mathcal{S}_i$.

**Graph Implicit Models:** Implicit models (Gu et al., 2020; Liu et al., 2021a; Zhong et al., 2024) typically follow the structure $Z_k = f(Z_{k-1}, X)$, where $f$ is a function, often parameterized by a neural network, $X$ represents the input data, and $Z$ denotes the learned representation. The fixed-point solution can be obtained iteratively as $Z^* = \lim_{k \to \infty} Z_{k+1} = \lim_{k \to \infty} f(Z_k, X) = f(Z^*, X)$. Finally, the implicit models output the fixed-point representation for the downstream tasks.

**Labeling Trick:** The labeling trick (Zhang et al., 2021) in subgraph learning is a technique aimed at enhancing the expressive power of Graph Neural Networks (GNNs) for multi-node representation learning. It involves assigning labels to nodes based on their membership in a subgraph. A common example is the zero-one labeling trick, where nodes belonging to the target subgraph are labeled as 1, and all others are labeled as 0. The zero-one labeling trick requires relabeling for each different target subgraph, which can be computationally expensive. To address this, a more efficient variant, the max-zero-one (Wang & Zhang, 2021) labeling trick, enables batch processing by assigning labels based on the presence of any subgraph in the batch. For instance, a node is labeled as 1 if it belongs to a subgraph in the batch, allowing for the handling of multiple target subgraphs in a single forward pass.

### 3.2. Enhanced Subgraph Representation

The state-of-the-art subgraph learning algorithms can be broadly categorized into two approaches: 1) learning from subgraphs, such as SubGNN (Alsentzer et al., 2020); and 2) learning from nodes, such as GLASS (Wang & Zhang, 2021) and SSNP (Jacob et al., 2023). As discussed earlier,

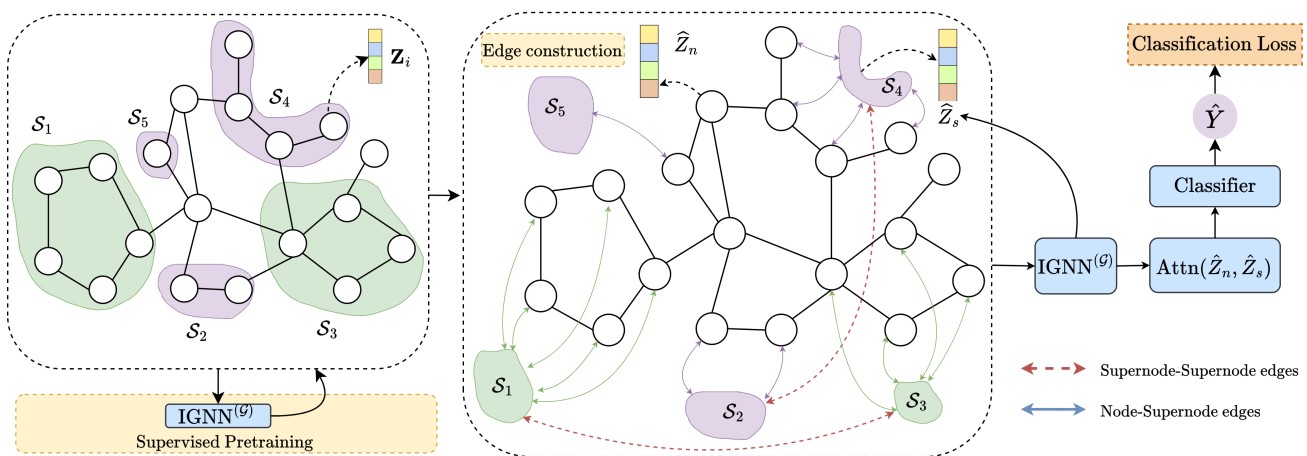

Figure 2: Overview of our method. We first construct a hybrid graph that consists of both regular nodes and supernodes, representing subgraphs. The edges between supernodes are inferred via pertaining (See Section 3.5). Finally, we learn node and subgraph embeddings jointly. The entire framework is trained using a novel bi-level optimization algorithm.

SubGNN does not fully leverage node-level information, and its subgraph-level information is not powerful enough to lead to state-of-the-art performance. On the other hand, methods like GLASS and SSNP completely ignore the subgraph information. However, it is intuitive that subgraph information, if leveraged properly, is essential in subgraph representation learning. Consider the example in Figure 1. The dependency between subgraphs $\mathcal{S}_1$ and $\mathcal{S}_3$ is difficult to capture in the node-level graph but is easy to establish in the subgraph-level graph.

To address these limitations, we propose a label-aware **subgraph enhancing framework** that integrates both subgraph-level and node-level information, offering flexibility and expressivity. The high-level concept is illustrated in Figure 2. The subgraph enhancing framework starts by constructing a hybrid graph $\hat{G} = \{\{\mathcal{V} \cup \mathcal{V}_s\}, \{\mathcal{E} \cup \mathcal{E}_s \cup \mathcal{E}_{ns}\}, \hat{X}\}$, where each **supernode** in $\mathcal{V}_s$ represents a subgraph. $\mathcal{E}_s$ is the set of edges among super-nodes in $\mathcal{V}_s$ (capturing subgraph-subgraph relationship), and $\mathcal{E}_{ns}$ is the set of edges between $\mathcal{V}$ and $\mathcal{V}_s$ (capturing membership relationship). $\hat{X} \in \mathbb{R}^{(n+m) \times d}$ is the corresponding feature matrix. Then, we obtain $\{\hat{Z}_n, \hat{Z}_s\}$ the embeddings of $\hat{G}$ using the proposed approach. Finally, the subgraph embeddings are obtained by using attention

$$Z = \text{Attention}(\hat{Z}_n, \hat{Z}_s)$$

We will discuss the exact $\hat{G}$ construction in a later section. However, as demonstrated by the following proposition, the subgraph enhancing framework is more flexible than the labeling trick proposed in (Wang & Zhang, 2021) (also described in the Preliminaries section).

**Proposition 3.1.** *Max-zero-one labeling trick is a special case of subgraph enhancing.*

*Proof.* Subgraph enhancement can simulate the labeling trick with an additional aggregation step and an auxiliary dimension. Subgraph nodes send messages to other nodes, and if a node in the node-level graph receives any message, its auxiliary feature is marked as 1; otherwise, it is marked as 0. Afterward, the subgraph-level graph can be disregarded. □

By incorporating appropriate subgraph-level information, our framework enhances the expressivity of GNNs, enabling them to distinguish between subgraphs that the max-zero-one labeling trick fails to differentiate, as shown in Figure 3. Consider a graph with no node features. In this case, $\mathcal{S}_1$ and $\mathcal{S}_2$ are indistinguishable, even with the max-zero-one labeling trick, since all nodes belong to at least one subgraph and are labeled 1. However, Figure 3 also demonstrates that by adding nodes to represent subgraphs and directed subgraph-level edges, the two trees become distinguishable.

Another key challenge in subgraph learning is that subgraphs of the same class may be widely separated within the base graph. Effectively learning meaningful embeddings for these subgraphs requires capturing long-range dependencies between them. Graph Implicit Models (GIMs) are particularly well-suited for this task, as they naturally propagate information across distant nodes. To address this issue, we propose the Implicit Subgraph Neural Network (ISNN). In the following sections, we formally introduce our model, provide theoretical guarantees, and detail the construction of the hybrid graph $\hat{G}$.

### 3.3. Formulation and Optimization

A straightforward way ahead is to train an implicit model directly on the hybrid graph $\hat{G}$. However, as demonstrated by

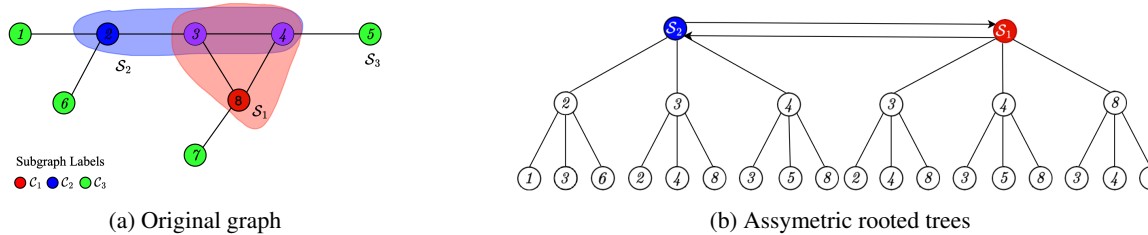

(a) Original graph  (b) Assymetric rooted trees

Figure 3: An example where max-zero-one labeling trick fails. Consider the graph on the left with empty features. GNN cannot distinguish $\mathcal{S}_1$ and $\mathcal{S}_2$ since every node in these subgraphs has the same topology, and the max-zero-one label trick assigns 1 to all nodes. The figure on the right demonstrates these two subgraphs can be distinguishable if the introduced subgraph-level edge has asymmetric weights.

our experiments, this approach does not lead to a desirable performance. This is primarily because directly computing fixed-point embeddings on the hybrid graph which involves a relatively complicated aggregation step is unstable (this is the main reason why standard IGNN baselines do not perform well for subgraph-level tasks). To overcome this issue, we formulate our problem as a bilevel optimization problem and solve it approximately. Our bi-level objective can be written as follows:

$$\min_{\boldsymbol{x} \in \mathcal{X}} F(\boldsymbol{x}; \mathbf{Z}^*) \triangleq \sum_{i=1}^{m} \ell(y_i, \phi_\theta([\mathbf{Z}^*]_\mathbf{i})) \tag{1}$$

$$\text{s.t. } \mathbf{Z}^* \triangleq \arg\min_{\mathbf{Z}} \frac{1}{2} \|\mathbf{Z} - f(\mathbf{Z}, \hat{G}; \xi, \mathbf{W})\|_F^2$$

where $\mathbf{Z}^* \in \mathbb{R}^{m \times d}$ is the subgraph embeddings. $[\mathbf{Z}^*]_\mathbf{i}$ denotes the $i$-th row of $\mathbf{Z}^*$. $\ell$ represents the classification loss, e.g., cross-entropy, and $\phi_\theta$ is a classifier parameterized by $\theta$. Let $\boldsymbol{x}$ denote the set of all parameters, and $\mathcal{X}$ denotes the feasible set. This paper adopts a linear fixed pooling function and simple mean aggregation. However, our approach is flexible and supports learnable pooling and aggregation functions. $f$ is an implicit model parameterized by $\xi$ and $\mathbf{W}$. In particular, we use the EIGNN (Liu et al., 2021a) formulation for $f$.

$$f(\mathbf{Z}, \hat{G}; \xi, \mathbf{W}) = \alpha \mathbf{A} \mathbf{Z} h(\mathbf{W}) + \psi_\xi(\hat{\mathbf{X}}) \tag{2}$$

$$h(\mathbf{W}) = \frac{\mathbf{W}^T \mathbf{W}}{\|\mathbf{W}\|\|\mathbf{W}\| + e_h} \tag{3}$$

In the equations presented above, $\mathbf{A}$ denotes the normalized adjacency matrix, while $\alpha \in (0, 1)$ is a controlling, and $e_h$ is a small float for numerical stability. The $\alpha$ is selected small enough so that $f$ is a non-expansive operator w.r.t. $\mathbf{Z}$.

We propose Algorithm 1 to find stationary solution of problem 1. The algorithm consists of two for-loops. The inner loop runs the fixed-point iteration $K$ times, and the outer loop runs gradient descent with proxy embeddings, where $\nabla f := (\partial_{\boldsymbol{x}} f, \partial_{\mathbf{Z}} f)$ and $\hat{\nabla} f := (\partial_{\boldsymbol{x}} f, 0)$. This training algorithm follows the framework of V-PBGD from (Shen &

Chen, 2023), which is a first-order Hessian-free penalty-based nonconvex bilevel optimization algorithm. Such a penalty-based algorithm solves an alternative problem

$$\min_{\boldsymbol{x} \in \mathcal{X}, \mathbf{Z}} F(\boldsymbol{x}, \mathbf{Z}) + \gamma \left(g(\boldsymbol{x}, \mathbf{Z}) - g(\boldsymbol{x}, \mathbf{Z}^*)\right) \tag{4}$$

where $g(\boldsymbol{x}, \mathbf{Z}) = \frac{1}{2}\|\mathbf{Z} - f(\mathbf{Z}, \hat{G}; \xi, \mathbf{W})\|_F^2$ denotes the lower-level problem. If $g$ is a Polyak-Łojasiewicz (PL) function, solving this problem with large enough $\gamma$ can deliver a stationary solution (Proposition 1 & 2 in (Shen & Chen, 2023)). Since solving $\mathbf{Z}^*$ for each gradient descent step is intractable, prior works use proxy variables to approximate it (Hu et al., 2022).

The main difference between our algorithm and V-PBGD is that we used the fixed-point iteration to update the proxy embeddings $\hat{\mathbf{Z}}$. We will show that our objective is a special case of nonconvex bilevel problem with lower-level PL function. Thus, this allows us to adopt existing techniques, and the additional non-expansiveness allows us to get a better gradient oracle complexity (see Remark 3.9). (Zhong et al., 2024) also proposed a bilevel algorithm using fixed-point iteration in the inner loop. However, their algorithm is heuristic and does not provide convergence guarantees.

### 3.4. Convergence Analysis

Let us start with some definitions.

**Definition 3.2.** A function $f$ is said to be $\mu$-contractive if it satisfies $\|f(x) - f(y)\|^2 \leq \mu\|x - y\|^2$, for any $x, y$ in the domain of $f$.

**Definition 3.3.** A function $f$ is said to be a $\mu$-Polyak-Łojasiewicz function if it satisfies $\|\nabla f(x)\|^2 \geq \mu(f(x) - f^*)$, for any $x$ in the domain of $f$ and $f^* = \inf f(x)$.

To state the convergence result, we first make the following assumptions.

**Assumption 3.4.** The upper-level function $F$ and lower-level function $g$ are $L_F, L_g$-smooth for all variables, the upper-level function $F$ is $l_F$-Lipschitz for all variables.

---

**Algorithm 1** ISNN Training Algorithm

---

1: **Input:** Graph $\hat{G} = (\mathcal{V} \cup \mathcal{V}_s, \mathcal{E} \cup \mathcal{E}_s \cup \mathcal{E}_{ns}, \hat{X})$, Learning rate $\eta$, hyperparameter $\gamma$;
2: $\mathbf{Z}^1 \leftarrow 0$;
3: **for** i=1,...,T **do**
4:     $\hat{\mathbf{Z}}_0^i \leftarrow \mathbf{Z}^i$
5:     **for** j=1,...,K **do**
6:         $\hat{\mathbf{Z}}_j^i \leftarrow f(\hat{\mathbf{Z}}_{j-1}^i, \hat{G};)$;
7:     **end for**
8:     $\nabla g := \nabla g(\boldsymbol{x}^i, \mathbf{Z}^i) - \bar{\nabla} g_k(\boldsymbol{x}^i, \hat{\mathbf{Z}}_K^i)$
9:     $\nabla F_\gamma := \nabla F(\boldsymbol{x}^i; \mathbf{Z}^i) + \gamma \nabla g$
10:    $(\boldsymbol{x}^{i+1}, \mathbf{Z}^{i+1}) \leftarrow \text{Proj}\left((\boldsymbol{x}^i, \mathbf{Z}^i) - \eta \nabla F_\gamma\right)$
11: **end for**
12: **Return:** $(\boldsymbol{x}^T, \mathbf{Z}^T)$.

---

**Assumption 3.5.** Assume that $f(\boldsymbol{x}, \cdot)$ is $\mu$-contractive for all $\boldsymbol{x} \in \mathcal{X}$ and $\mathcal{X}$ is nonempty.

Note that the Assumption 3.4 is standard in bilevel optimization literature (Hu et al., 2022; Shen & Chen, 2023). The additional Assumption 3.5 is a natrual requirement for implicit models to ensure the well-posedness (Gu et al., 2020), which we can maintain by choosing small enough $\alpha$. We defer most of the proofs in this section to the Appendix.

*Remark* 3.6. The lower-level problems $g$ is a $2(1-\mu)$ PL function.

*Proof.* For simplicity, we denote $g(\boldsymbol{x}, \mathbf{Z}), f(\mathbf{Z}, \hat{G}; \xi, \mathbf{W})$ by $g_x(\mathbf{Z}), f_x(Z)$, and $\vec{\mathbf{Z}}$ denotes the vectorized $\mathbf{Z}$. By the definition of function $g$, we have

$$\nabla g_x(\vec{\mathbf{Z}}) = (I - \frac{\partial f_x(\vec{\mathbf{Z}})}{\vec{\mathbf{Z}}})(\vec{\mathbf{Z}} - f_x(\vec{\mathbf{Z}}))$$

$$\Rightarrow \|\nabla g_x(\vec{\mathbf{Z}})\|^2 = \|(I - \frac{\partial f_x(\vec{\mathbf{Z}})}{\vec{\mathbf{Z}}})(\vec{\mathbf{Z}} - f_x(\vec{\mathbf{Z}}))\|^2 \quad (5)$$

Since $f_x$ is a $\mu$-contractive function, the operator norm of its Jacobian is bounded by $\mu$. Therefore, the operator norm of $I - \frac{\partial f_x(\vec{\mathbf{Z}})}{\vec{\mathbf{Z}}}$ is lower bounded by $1 - \mu$. Thus,

$$\|\nabla g_x(\mathbf{Z})\|^2 \geq 2(1-\mu) g_x(\mathbf{Z}) \quad (6)$$

Moreover,

$$\|\nabla g_x(\vec{\mathbf{Z}})\|^2 = \|(I - \frac{\partial f_x(\vec{\mathbf{Z}})}{\vec{\mathbf{Z}}})(\vec{\mathbf{Z}} - f_x(\vec{\mathbf{Z}}))\|^2 \leq 2 g_x(\mathbf{Z}) \quad (7)$$

$\square$

*Remark* 3.7. Based on the Assumption 3.5, the model is well-posed i.e., the lower-level problem admits a unique fixed point.

Now, we are ready to state our convergence result.

**Theorem 3.8.** *Consider Algorithm 1, suppose Assumption 3.5 and 3.4 hold. Choosing $\eta \in (0, L_F + \gamma(2L_g + \frac{L_g^2}{2(1-\mu)})]$, $K = \Omega(\log \frac{\sqrt{m}\eta t}{1-\mu})$, and $\gamma = \Omega(\epsilon^{-0.5})$, where $\epsilon$ is the desired error. Let $\boldsymbol{z}$ denote the list of all parameters $\{\boldsymbol{x}, \mathbf{Z}\}$, and $A_\gamma(\boldsymbol{z}) := F(\boldsymbol{x}, \mathbf{Z}) + \gamma(g(\boldsymbol{x}, \mathbf{Z}) - g(\boldsymbol{x}, \mathbf{Z}^*))$ be the loss of penlty problem. Let $\bar{\boldsymbol{z}}^{t+1} := Proj(\boldsymbol{z}^t - \nabla A_\gamma(\boldsymbol{z}^t))$ be the solution updated by the exact gradient. We have the following*

$$\sum_{t=1}^T \|\bar{\boldsymbol{z}}^{t+1} - \boldsymbol{z}^t\| \leq \frac{18 A_\gamma(\boldsymbol{z}^1)}{\eta} + 10 l_F^2 L_g^2 \quad (8)$$

*Remark* 3.9. Theorem 3.8 indicates the Algorithm 1 can produce a stationary solution to the bilevel optimization problem if it converges. The sample complexity is $\tilde{O}(\epsilon^{-1.5})$, which matches the result in (Shen & Chen, 2023). To note that, our method only computes gradient once per iteration. However, the method in (Shen & Chen, 2023) needs to compute $K + 1$ gradients per iteration. Thus, our method is more efficient in training implicit models.

### 3.5. Hybrid Graph Construction

There are several elements in the hybrid graph $\hat{G}$ that need to be described: $\mathcal{E}_s, \mathcal{E}_{ns}$, and $\hat{X}$. The feature matrix $\hat{X}$ can be constructed by concatenating node features with subgraph features. The subgraph feature, in turn, can be defined as the average node feature within each subgraph. Similarly, an edge $(i, j) \in \mathcal{E}_{ns}$ exists if node $i$ is a member of subgraph $j$. The key challenge is defining $\mathcal{E}_s$. This paper presents three possible ways to define it. However, before diving into these details, let us first explore alternative methods.

Many subgraph-level methods (e.g., Sub2Vec and SubGNN) adopt a similar framework, but they learn embeddings solely from the subgraph-level graph, i.e., $\hat{G} = \{\mathcal{V}_s, \mathcal{E}_s, \hat{X}\}$. These methods construct $\mathcal{E}_s$ using three approaches – *1) Position*: defines edge weights based on the distance between subgraphs, *2) Neighborhood*: Computes edge weights using the number of overlapping neighbors between subgraphs, and *3) Structure*: Measures edge weights based on the topological similarity of subgraphs. However, these approaches do not achieve good performance in practice. We empirically show that they perform no better than assigning random edge weights (see Section 4.2). This is because these strategies ignore the downstream tasks, and the three types of information may not necessarily be beneficial for later training.

Therefore, instead of focusing on the structural properties of the graph, we propose an approach that directly addresses the task. For a classification task, two intuitive assumptions can be made: 1) Embeddings of supernodes belonging to the same class should be similar. 2) A test supernode is more

Table 1: Statistic of datasets.

| Dataset | #Nodes | #Edges | #Subgraphs | #Labels/Classes |
|---------|--------|--------|------------|-----------------|
| PPI-BP | 17,080 | 316,591 | 1,591 | 6 |
| HPO-METAB | 14,587 | 3,238,174 | 2,400 | 6 |
| HPO-NEURO | 14,587 | 3,238,174 | 4,000 | 10 |
| EM-USER | 57,333 | 4,573,417 | 324 | 2 |

likely to share the same label as its most similar supernode in the training set.

To this end, our framework begins by pretraining the model with an empty $\mathcal{E}_s$. Based on the pretrained embeddings and these two assumptions, we introduce two label-aware strategies tailored for different scenarios:

- *Binary & Multi-class*: For each class, we connect $k$ pairs of the most distant training supernodes.
- *Multi-label*: We select certain labels and convert them from binary to integer values, thereby treating the data as multi-class.

In all these strategies, each test node is connected to its closest training node in the latent space during inference.

# 4. Experiments

**Dataset:** We assess our model's performance and scalability by benchmarking it against various subgraph classification baselines on four real-world datasets. Following the experimental setup of SubGNN (Alsentzer et al., 2020), we evaluate our approach on *PPI-BP*, *HPO-METAB*, *HPO-NEURO*, and *EM-USER*. The statistics of the datasets are summarized in Table 1.

**Models:** We take the following models as our baselines.

- **Plain/Soft models**: use the backbone model to generate node embeddings then aggregate them to subgraph embeddings. The soft models are the plain models trained by our optimization algorithm.

- **Sub2Vec**: generates subgraph embeddings by first capturing two main properties, neighborhood and structural, of each subgraph and projecting these features into a low-dimensional continuous vector space.

- **GLASS**: employs a max-zero-one labeling trick, distinguishing nodes within the target subgraph from those outside it. GLASS effectively captures the internal and external topology that influence subgraph properties using a standard message passing framework.

- **SubGNN**: propagates neural messages between subgraph components and anchor patches from the underlying graph by utilizing three distinct channels: neighborhood, structure, and position to capture various as-

Table 2: Micro-F1 on real-world datasets averaged over 10 runs, for all models. **Bold** indicates the best result, and underline indicates the second best result.

| Method | PPI-BP | HPO-METAB | HPO-NEURO | EM-USER |
|--------|--------|-----------|-----------|---------|
| MLP | $0.297_{\pm 0.027}$ | $0.443_{\pm 0.063}$ | $0.490_{\pm 0.059}$ | $0.808_{\pm 0.138}$ |
| GCN-plain | $0.398_{\pm 0.058}$ | $0.452_{\pm 0.025}$ | $0.535_{\pm 0.032}$ | $0.561_{\pm 0.021}$ |
| Sub2Vec | $0.309_{\pm 0.023}$ | $0.114_{\pm 0.021}$ | $0.206_{\pm 0.073}$ | $0.522_{\pm 0.043}$ |
| GLASS | $0.618_{\pm 0.006}$ | $\underline{0.598_{\pm 0.014}}$ | $0.675_{\pm 0.007}$ | $0.884_{\pm 0.008}$ |
| SubGNN | $0.598_{\pm 0.032}$ | $0.531_{\pm 0.015}$ | $0.644_{\pm 0.009}$ | $0.815_{\pm 0.054}$ |
| SSNP | $\underline{0.636_{\pm 0.007}}$ | $0.587_{\pm 0.010}$ | $\underline{0.682_{\pm 0.004}}$ | $\underline{0.888_{\pm 0.005}}$ |
| IGNN-plain | $0.389_{\pm 0.025}$ | $0.284_{\pm 0.021}$ | $0.215_{\pm 0.002}$ | $0.579_{\pm 0.008}$ |
| EIGNN-plain | $0.425_{\pm 0.050}$ | $0.252_{\pm 0.009}$ | $0.312_{\pm 0.017}$ | $0.591_{\pm 0.006}$ |
| SoftIGNN | $0.594_{\pm 0.006}$ | $0.520_{\pm 0.002}$ | $0.653_{\pm 0.005}$ | $0.820_{\pm 0.008}$ |
| SoftEIGNN | $0.592_{\pm 0.006}$ | $0.522_{\pm 0.002}$ | $0.658_{\pm 0.004}$ | $0.829_{\pm 0.010}$ |
| ISNN | $\mathbf{0.731_{\pm 0.026}}$ | $\mathbf{0.646_{\pm 0.014}}$ | $\mathbf{0.688_{\pm 0.004}}$ | $\mathbf{0.914_{\pm 0.009}}$ |

Table 3: AUROC scores on real-world datasets averaged over 10 runs, for all models. **Bold** indicates the best result, and underline indicates the second best result.

| Method | PPI-BP | HPO-METAB | HPO-NEURO | EM-USER |
|--------|--------|-----------|-----------|---------|
| MLP | $0.498_{\pm 0.009}$ | $0.814_{\pm 0.032}$ | $0.764_{\pm 0.104}$ | $0.896_{\pm 0.143}$ |
| GCN-plain | $0.663_{\pm 0.044}$ | $0.772_{\pm 0.018}$ | $0.773_{\pm 0.027}$ | $0.525_{\pm 0.065}$ |
| Sub2Vec | $0.544_{\pm 0.011}$ | $0.496_{\pm 0.010}$ | $0.504_{\pm 0.015}$ | $0.518_{\pm 0.048}$ |
| GLASS | $\underline{0.835_{\pm 0.002}}$ | $\underline{0.891_{\pm 0.002}}$ | $0.852_{\pm 0.001}$ | $\mathbf{0.960_{\pm 0.004}}$ |
| SubGNN | $0.816_{\pm 0.012}$ | $0.862_{\pm 0.005}$ | $0.843_{\pm 0.014}$ | $0.911_{\pm 0.042}$ |
| SSNP | $0.831_{\pm 0.008}$ | $0.883_{\pm 0.007}$ | $0.867_{\pm 0.004}$ | $0.952_{\pm 0.011}$ |
| IGNN-plain | $0.514_{\pm 0.046}$ | $0.496_{\pm 0.063}$ | $0.709_{\pm 0.065}$ | $0.541_{\pm 0.089}$ |
| EIGNN-plain | $0.630_{\pm 0.189}$ | $0.579_{\pm 0.092}$ | $0.601_{\pm 0.121}$ | $0.553_{\pm 0.072}$ |
| SoftIGNN | $0.797_{\pm 0.005}$ | $0.818_{\pm 0.001}$ | $\underline{0.868_{\pm 0.004}}$ | $0.932_{\pm 0.005}$ |
| SoftEIGNN | $0.798_{\pm 0.008}$ | $0.821_{\pm 0.001}$ | $\underline{0.868_{\pm 0.002}}$ | $0.927_{\pm 0.006}$ |
| ISNN | $\mathbf{0.924_{\pm 0.012}}$ | $\mathbf{0.919_{\pm 0.002}}$ | $\mathbf{0.896_{\pm 0.002}}$ | $\underline{0.959_{\pm 0.005}}$ |

pects of subgraph topology. This enables SubGNN to have expressive subgraph representations.

- **SSNP**: aggregates both subgraph and neighborhood information without any labeling tricks to classify subgraphs. SSNP generates subgraph embeddings by first processing node features using transformation layers (GCN, MLP, etc.) and then, for every subgraph, aggregating the node features of the subgraph and its neighborhood through SSNP.

**Hyperparameter:** For all datasets, we employ a 2-layer GNN and a 2-layer MLP for classification, with a fixed hidden dimension of 64, consistent with the SubGNN settings. For our method, we vary the value of $\gamma$ in $\{0.0001, 0.001, 0.01\}$ and the number of inner-loop iterations $K$ in $\{1, 2, 3, 4, 5\}$.

**Configuration:** We rerun each experiment 10 times (max of 1500 epochs) and report the average performance. We conducted experiments on AMD EPYC 7763 64-Core Processor with 2 TB memory and on 8 NVIDIA A30 GPUs.

## 4.1. Subgraph Classification

The results are presented in Tables 2 and 3, demonstrating that our model outperforms baselines across almost all

Table 4: Performance of four variants of our model, which uses different ways of constructing subgraph-level graph.

| HPO-METAB | ISNN-P | ISNN-S | IGNN-N | ISNN-rand |
|---|---|---|---|---|
| F1 | $0.586_{\pm 0.021}$ | $0.598_{\pm 0.021}$ | $0.595_{\pm 0.021}$ | $0.589_{\pm 0.028}$ |
| AUROC | $0.874_{\pm 0.007}$ | $0.874_{\pm 0.010}$ | $0.876_{\pm 0.007}$ | $0.876_{\pm 0.008}$ |

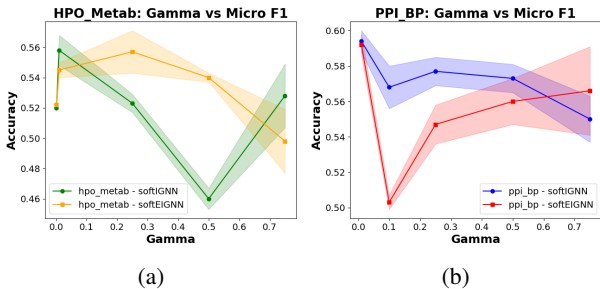

(a)    (b)

Figure 4: Sensitivity analysis on $\gamma$. Running SoftIGNN and SoftEIGNN on HPO-METAB and PPI-BP datasets.

Table 5: Average runtime per run of 1500 epochs in seconds over 10 runs

| Method | PPI-BP | HPO-NEURO | HPO-METAB | EM-USER |
|---|---|---|---|---|
| SSNP | $130.47_{\pm 4.120}$ | $204.15_{\pm 25.78}$ | $162.34_{\pm 19.45}$ | $158.29_{\pm 28.33}$ |
| IGNN-plain | $439.29_{\pm 58.74}$ | $1629.86_{\pm 89.14}$ | $1142.88_{\pm 97.42}$ | $1386.28_{\pm 85.90}$ |
| EIGNN-plain | $114.35_{\pm 0.237}$ | $275.48_{\pm 1.489}$ | $185.82_{\pm 0.775}$ | $176.99_{\pm 5.405}$ |
| ISNN | $\mathbf{104.66}_{\pm 28.14}$ | $\mathbf{128.26}_{\pm 4.571}$ | $\mathbf{160.83}_{\pm 19.37}$ | $\mathbf{135.29}_{\pm 35.70}$ |

datasets and metrics. Notably, our model excels on the two multi-class datasets, PPI-BP and HPO-METAB, where label-aware subgraph information plays a crucial role. By leveraging this information, the model learns more concentrated embeddings for each class, resulting in a clearer classification boundary. Even in the simpler binary classification setting (EM-USER), our approach achieves an approximately 3% improvement in F1 score compared to the state-of-the-art method. However, for the multi-label dataset HPO-NEURO, the conversion process for class labels does not fully capture the relationships between subgraphs, limiting our advantage.

We also observe that training GNNs or implicit models naively for subgraph representation learning is challenging and unstable. However, our bilevel optimization algorithm enables stable training of implicit models while achieving performance comparable to `SubGNN`. Notably, `SubGNN` leverages subgraph-level graphs to learn embeddings, effectively providing shortcuts in the original graph. The fact that soft implicit models achieve similar performance suggests that it successfully captures long-range dependencies without relying on subgraph-level information, as intended.

### 4.2. Ablation Study

**Subgraph-level Information:** In this section, we investigate whether subgraph-level information can aid classification and, if so, which types are most effective. We conduct experiments on HPO-METAB, considering four methods for constructing subgraph-level graphs: *position*, *neighborhood*, *structure*, and *random*. The first three methods have been introduced in Section 3.5, while the *random* method assigns edge weights following a normal distribution.

Based on these constructions, we define four model variants: **ISNN-P**, **ISNN-N**, **ISNN-S**, and **ISNN-rand**, corresponding to the *position*, *neighborhood*, *structure*, and *random* subgraph-level information, respectively. The results are summarized in Table 4. While all models achieve performance comparable to state-of-the-art methods, the choice of subgraph construction does not appear to significantly impact the results. Surprisingly, all variants perform similarly, including the *random* approach.

**Sensitivity:** In our optimization algorithm, the trade-off parameter $\gamma$ in Equation (4) plays a crucial role in controlling the strength of the fixed-point constraint. In this section, we

perform a sensitivity analysis on $\gamma$ to evaluate its impact on our algorithm. We conduct experiments by running SoftIGNN and SoftEIGNN on HPO-METAB and PPI-BP with varying $\gamma$. The results are presented in Figure 4.

As observed, both methods exhibit poorer performance and higher variance as $\gamma$ increases. A larger $\gamma$ implies that the soft models become more similar to conventional models. This experiment suggests that strictly enforcing the fixed-point constraint may negatively impact the training process.

### 4.3. Efficiency

We also measure the runtime of our method compared to baseline approaches. The results, presented in Table 5, confirm that our algorithm introduces minimal overhead. This is expected, as the inner loop consists solely of fixed-point iterations, which do not require gradient computation.

## 5. Conclusion

We introduced `ISNN`, the first implicit model for subgraph representation learning, along with a provably convergent bilevel optimization algorithm for training. Additionally, we proposed a graph-enhancing framework to learn more expressive subgraph representations. This framework consists of three steps: 1) Pretrain the model to obtain subgraph embeddings. 2) Construct a hybrid graph using the pretrained embeddings and label information. 3) Use the hybrid graph for subsequent training. As demonstrated in our experiments, this framework leads to significant improvements.

## Acknowledgement

This work was supported in part by the NSF Cyber-training award 2320980, NSF SCH award 2306331, a start-up from the University of Iowa, and the CDC under cooperative agreement U01-CK000594.

## Impact Statement

This paper presents work whose goal is to advance the field of Machine Learning. There are many potential societal consequences of our work, none of which we feel must be specifically highlighted here.

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

## A. Ablation Study on `SubGNN`

Additionally, we conduct an experiment where we replace all subgraph information of *position*, *neighborhood*, and *structure* used by `SubGNN` with *random*. The results, presented in Table 6, follow a similar trend.

Table 6: Average F1 score of `SubGNN` with *random* subgraph-level graph.

| Method | PPI-BP | HPO-METAB | HPO-NEURO | EM-USER |
|---|---|---|---|---|
| **SubGNN** | 0.598 | 0.531 | 0.644 | 0.815 |
| **SubGNN-rand** | 0.597 | 0.538 | 0.629 | 0.810 |

## B. Sensitivity to Subgraph-Level Graph Construction Methods

In our experiments, we previously compared four subgraph-level graph construction methods—*random*, *neighborhood*, *position*, and *structure*. The goal was to show that these classical methods do not significantly improve classification performance, as they achieve similar results to using random subgraph information. This observation echoes conclusions from prior work such as GRASS and SSNP.

To further the comparison, we introduce additional label-aware methods:

- **class-rand**: For each class, we randomly connect $k$ pairs of training supernodes.

- **Star**: For each class, we select a centroid supernode and connect it with all other supernodes within that class.

**Experimental Results on HPO-METAB**

| Metric | ISNN-rand | ISNN-class-rand | ISNN-Star | ISNN |
|---|---|---|---|---|
| F1 | 0.589 | 0.672 | 0.707 | 0.731 |
| AUROC | 0.876 | 0.901 | 0.916 | 0.924 |

Table 7: Performance comparison of different subgraph-level graph construction methods on HPO-METAB.

The `ISNN-class-rand` method improves performance over `ISNN-rand` by using label information to ensure that subgraphs within the same class have similar embeddings, making classification easier. The `ISNN-Star` method further boosts performance by selecting a centroid supernode for each class and connecting it to all other supernodes, reinforcing intra-class similarity, though it introduces more edges. Our `ISNN` method connects only the top-$k$ most distant pairs of supernodes, balancing label-aware benefits with a sparse graph structure.

## C. Additional Hyperparameter Sensitivity Analysis

We further investigate the sensitivity of the hyperparameter $\gamma$ on ISNN. The results are presented as follows

| Gamma | F1 (10 Runs) |
|---|---|
| 0.001 | $0.713 \pm 0.014$ |
| 0.01 | $0.726 \pm 0.022$ |
| 0.05 | $0.670 \pm 0.008$ |
| 0.1 | $0.655 \pm 0.007$ |
| 0.5 | $0.654 \pm 0.022$ |
| 1 | $0.634 \pm 0.025$ |

Table 8: F1 scores for different values of $\gamma$ on PPI_BP dataset.

| Gamma | F1 (10 Runs) |
|-------|-------------------|
| 0.001 | $0.869 \pm 0.012$ |
| 0.01  | $0.808 \pm 0.027$ |
| 0.05  | $0.853 \pm 0.027$ |
| 0.1   | $0.913 \pm 0.031$ |
| 0.5   | $0.857 \pm 0.018$ |
| 1     | $0.771 \pm 0.035$ |

Table 9: F1 scores for different values of $\gamma$ on EM_USER dataset.

These results show that the best performance occurs when $\gamma$ is between $0.01$ and $0.1$. For example, PPI_BP achieves its best performance at $\gamma = 0.01$, while EM_USER peaks at $\gamma = 0.1$. Gamma values outside this range lead to worse performance, highlighting the importance of using a moderate $\gamma$.

## D. Scalability of Our Method

The complexity of a standard GCN is given by:

$$O(n \cdot d^2 + E \cdot d) \tag{9}$$

where:

- $n$ is the number of nodes,
- $E$ is the number of edges, and
- $d$ is the hidden dimension.

For our method, the complexity becomes:

$$O((n + s) \cdot d^2 + (E + k) \cdot d) \tag{10}$$

where:

- $s$ is the number of subgraphs, and
- $k$ is the number of additional edges introduced.

Thus, the additional overhead introduced by ISNN is an additive term of:

$$O(s \cdot d^2 + k \cdot d) \tag{11}$$

Empirically, $k$ can be controlled to remain small, and $s$, which is part of the input, is typically small in most practical subgraph learning tasks. Therefore, ISNN exhibits scalability similar to that of a standard GCN in most practical scenarios.

## E. Proof of Theorem 3.8

Based on the analysis framework from (Shen & Chen, 2023), we present the proof of Theorem 3.8. Fisrt, we provide the convergence result on the inner loop.

**Lemma E.1.** *(Convergence of fixed point) Suppose Assumption 3.5 and 3.4 hold. Given any $\boldsymbol{x}^t \in \mathcal{X}$ and $\hat{\mathbf{Z}}_1^t$, after running $K$ steps of the inner updates, the $\hat{Z}_K^t$ satisfies*

$$\|\hat{\mathbf{Z}}_K^t - \mathbf{Z}^*(\boldsymbol{x}^t)\|^2 \le \frac{2\mu^K}{1 - \mu} g(\boldsymbol{x}^t, \hat{\mathbf{Z}}_0^t)$$

*Proof.* The inner loop updates: $\hat{\mathbf{Z}}_k^t = h(\boldsymbol{x}^t, \hat{\mathbf{Z}}_1^t)$. By assumption, we know $h(\cdot)$ is a $\mu$-contractive mapping

$$\|\hat{\mathbf{Z}}_K^t - h(\boldsymbol{x}^t, \hat{\mathbf{Z}}_K^t)\|^2 \leq \mu\|\hat{\mathbf{Z}}_{K-1}^t - h(\boldsymbol{x}^t, \hat{\mathbf{Z}}_{K-1}^t)\|^2 \leq \mu^K\|\hat{\mathbf{Z}}_0^t - h(\boldsymbol{x}^t, \hat{\mathbf{Z}}_0^t)\|^2$$

Moreover, we know $\|\mathbf{Z} - \mathbf{Z}^*(\boldsymbol{x}^t)\| \leq \frac{1}{1-\mu}\|\mathbf{Z} - h(\boldsymbol{x}^t, \mathbf{Z})\|$ for any $\mathbf{Z}$. Then, taking $\mathbf{Z}$ to be $\hat{\mathbf{Z}}_K^t$

$$\|\hat{\mathbf{Z}}_K^t - \mathbf{Z}^*(\boldsymbol{x}^t)\|^2 \leq \frac{1}{1-\mu}\|\hat{\mathbf{Z}}_K^t - h(\boldsymbol{x}^t, \hat{\mathbf{Z}}_K^t)\|^2 \leq \frac{\mu^K}{1-\mu}\|\hat{\mathbf{Z}}_0^t - h(\boldsymbol{x}^t, \hat{\mathbf{Z}}_0^t)\|^2 = \frac{2\mu^K}{1-\mu}g(\boldsymbol{x}^t, \hat{\mathbf{Z}}_0^t) \tag{12}$$

$\square$

Next, we will give a proof to Theorem 3.8.

*Proof.* (of Theorem 3.8) In this proof, we denote $\boldsymbol{z} = (\boldsymbol{x}, \mathbf{Z})$. Then the update can be written as

$$\boldsymbol{z}^{t+1} = \text{Proj}(\boldsymbol{z}^t - \eta\nabla A_\gamma(\boldsymbol{x}^t, \hat{\mathbf{Z}}_K^t))$$

where $\hat{\nabla} A_\gamma(\boldsymbol{z}^t, \hat{\mathbf{Z}}_K^t) := \nabla F(\boldsymbol{z}^t) + \gamma(\nabla g(\boldsymbol{z}^t) - \bar{\nabla}g(\boldsymbol{x}^t, \hat{\mathbf{Z}}_K^t))$. Based on the assumptions we made and Lemma A.5 from (Nouiehed et al., 2019), function $A_\gamma$ is $L_\gamma$-Lipschitz-smooth with $L_\gamma = L_F + \gamma(2 + \mu)$. Then the smoothness implies

$$\begin{aligned} A_\gamma(\boldsymbol{z}^{t+1}) \leq & A_\gamma(\boldsymbol{z}^t) + \langle\nabla A_\gamma(\boldsymbol{z}^t), \boldsymbol{z}^{t+1} - \boldsymbol{z}^t\rangle + \frac{L_\gamma}{2}\|\boldsymbol{z}^{t+1} - \boldsymbol{z}^t\|^2 \\ \leq & A_\gamma(\boldsymbol{z}^t) + \underbrace{\langle\hat{\nabla} A_\gamma(\boldsymbol{z}^t, \hat{\mathbf{Z}}_K^t), \boldsymbol{z}^{t+1} - \boldsymbol{z}^t\rangle}_{\spadesuit} + \frac{1}{2\eta}\|\boldsymbol{z}^{t+1} - \boldsymbol{z}^t\|^2 \\ & + \underbrace{\langle\nabla A_\gamma(\boldsymbol{z}^t) - \hat{\nabla} A_\gamma(\boldsymbol{z}^t, \hat{\mathbf{Z}}_K^t), \boldsymbol{z}^{t+1} - \boldsymbol{z}^t\rangle}_{\clubsuit} \end{aligned} \tag{13}$$

The second inequality is because we ensure $\eta \leq \frac{1}{L_\gamma}$. Now the problem is bounding terms $\clubsuit, \spadesuit$.

For $\spadesuit$: the $\boldsymbol{z}^{t+1}$ can be represented as the minimizer of the following problem

$$\min_{\boldsymbol{z}\in\mathcal{Z}}\langle\hat{\nabla} A_\gamma(\boldsymbol{z}^t, \hat{\mathbf{Z}}_K^t), \boldsymbol{z}\rangle + \frac{1}{2\eta}\|\boldsymbol{z} - \boldsymbol{z}^t\|^2$$

Then, according to the first order optimality,

$$\begin{aligned} & \langle\hat{\nabla} A_\gamma(\boldsymbol{z}^t, \hat{\mathbf{Z}}_K^t) + \frac{1}{\eta}(\boldsymbol{z}^{t+1} - \boldsymbol{z}^t), \boldsymbol{z} - \boldsymbol{z}^t\rangle \leq 0, \forall \boldsymbol{z} \in \mathcal{Z} \\ \Rightarrow & \langle\hat{\nabla} A_\gamma(\boldsymbol{z}^t, \hat{\mathbf{Z}}_K^t), \boldsymbol{z}^{t+1} - \boldsymbol{z}^t\rangle \leq -\frac{1}{\eta}\|(\boldsymbol{z}^{t+1} - \boldsymbol{z}^t)\|^2 \end{aligned} \tag{14}$$

Consider $\clubsuit$, by Young's inequality, we have

$$\langle\nabla A_\gamma(\boldsymbol{z}^t) - \nabla A_\gamma(\boldsymbol{z}^t, \hat{\mathbf{Z}}_K^t), \boldsymbol{z}^{t+1} - \boldsymbol{z}^t\rangle \leq \eta\underbrace{\|\nabla A_\gamma(\boldsymbol{z}^t) - \nabla A_\gamma(\boldsymbol{z}^t, \hat{\mathbf{Z}}_K^t)\|^2}_{\star} + \frac{1}{4\eta}\|\boldsymbol{z}^{t+1} - \boldsymbol{z}^t\|^2 \tag{15}$$

then

$$
\begin{aligned}
\bigstar &= \gamma^2 \|\nabla g(\boldsymbol{x^t}, \mathbf{Z}^*) - \nabla g(\boldsymbol{x^t}, \hat{\mathbf{Z}}_K^t)\|^2 \\
&\leq \gamma^2 L_g^2 \|\mathbf{Z}^* - \hat{\mathbf{Z}}_K^t\|^2 \quad \text{(By Assumption 3.4)} \\
&\leq \frac{2\gamma^2 L_g^2 \mu^K}{1-\mu} g_0(\boldsymbol{x^t}, \hat{\mathbf{Z}}_0^t) \quad \text{(By Lemma E.1)} \\
&\leq \frac{\gamma^2 L_g^2 \mu^K}{(1-\mu)^2} \|\nabla_2 g(\boldsymbol{x^t}, \hat{\mathbf{Z}}_0^t)\|^2 \quad \text{(By Remark 3.6)} \\
&\leq \frac{\gamma^2 L_g^2 \mu^K}{(1-\mu)^2} \left( \frac{2}{\eta^2 \gamma^2} \|\mathbf{Z}^{t+1} - \mathbf{Z}^t\|^2 + \frac{2l_F^2}{\gamma^2} \right) \\
&\leq \frac{L_g^2 \mu^K}{(1-\mu)^2} \left( \frac{2}{\eta^2} \|\boldsymbol{z}^{t+1} - \boldsymbol{z}^t\|^2 + 2l_F^2 \right) \\
&\leq \frac{1}{8\eta^2} \|\boldsymbol{z}^{t+1} - \boldsymbol{z}^t\|^2 + \frac{l_F^2 L_g^2}{2\eta^2 t^2}
\end{aligned}
\tag{16}
$$

The last inequality is because we choose $K = \max\{-2\log_\mu \frac{\sqrt{2}L_g}{1-\mu}, -2\log_\mu \frac{\sqrt{2}\eta t}{1-\mu}\}$. Follow the steps after (B.9) in (Shen & Chen, 2023), we can obtain the bound.

$$
\sum_{t=1}^{T} \|\bar{\boldsymbol{z}}^{t+1} - \boldsymbol{z}^t\| \leq \frac{18 A_\gamma(\boldsymbol{z}^1)}{\eta} + 10 l_F^2 L_g^2
\tag{17}
$$

$\square$

