# OpenReview forum: "Implicit Subgraph Neural Network"
_ICML.cc/2025/Conference — ICML 2025 poster_

### Official Review · Reviewer_JyUo · 2025-03-09

**Overall Recommendation:** 4

**Summary:**

This paper proposes a bi-level optimization framework for subgraph-level predictive tasks, where the outer level is to minimize the subgraph-level prediction loss, and the inner-level is to enforce the fixed-point conditions of the implicit representation of the subgraphs, so that they don't have to rely on rigid subgraph definitions.

**Claims And Evidence:**

Yes

**Essential References Not Discussed:**

Could you please compare with GEQ (Efficient and scalable implicit graph neural networks with virtual equilibrium, https://drive.google.com/file/d/1u2zJ_LJyIEFOjatUiT2gG1QXVLkTY3_y/view?pli=1)?
It is not for subgraph learning tasks, but it also formulates node classification as a bilevel optimization problem and develops a provably convergent algorithm and "replace gradient descent in the inner loop with fixed-point iteration". Can this framework be adapted to subgraph learning also?

**Experimental Designs Or Analyses:**

I checked the comparison on subgraph classification datasets, and also the ablation studies about the subgraph-level Information and sensitivity analysis. Both looks good to me except the Sec 4.3 about the efficiency analysis.
As far as I know, EIGNN takes a nonnegligible amount of time for preprocessing. Could you please also specify the time consumption, besides average runtime per epoch?

**Methods And Evaluation Criteria:**

Yes.

**Other Comments Or Suggestions:**

N/A

**Other Strengths And Weaknesses:**

N/A

**Questions For Authors:**

Could you please resolve my questions in the Experimental Designs Or Analyses and Essential References Not Discussed?

**Relation To Broader Scientific Literature:**

This paper is a direct application of graph implicit models and bilevel optimization, where the latter enables the usage of the former to flexibly deal with the subgraph representation learning problem. This is a novel strategy since it is the first work to apply GIM to subgraph learning and benefits from the long-range dependencies and flexible problem design.

**Theoretical Claims:**

I checked the assumptions and conclusions of the theoretical claims about all the theorems. The assumption to ensure convergence is the general assumption made in implicit GNNs. I didn't check the detailed proof of Thm. 3.8, but the conclusion intuitively makes sense.

---

> ### Author Rebuttal · Authors · 2025-04-01
>
> We appreciate the reviewer's feedback.
>
> ---
>
> ### Time Consumption
>
> Yes, EIGNN requires a preprocessing step. Our method incorporates a pretraining stage. Below is a comparison of the total runtime (in seconds) for both methods on the PPI_BP dataset. The reported values are the means over 10 runs. We will update the running time results in the paper to reflect your suggestion.
>
> | Method | Preprocessing Time (s) | Training Time (s) | Total Time (s) |
> |--------|------------------------|-------------------|----------------|
> | ISNN  | 248.43                 | 807.69            | 1056.12        |
> | EIGNN   | 1201.84                | 956.54            | 2158.39        |
>
> ---
>
> ### Compare with GEQ
>
> Thanks for pointing out a missing reference. I think this paper proposes an interesting algorithm and has the potential to be adapted to subgraph representation learning. We would love to include this method as one of our baselines. However, the author did not release their code (the GitHub link does not work in their paper). We will reach out to them after the review period and include the paper in our literature review.

---

### Official Review · Reviewer_Red8 · 2025-03-11

**Overall Recommendation:** 3

**Summary:**

This paper introduced ISNN, the first implicit model for subgraph representation learning, along with a provably convergent bilevel optimization algorithm for training. The proposed ISNN also integrates label-aware subgraph-level information. This paper converts the fixed-point iteration into bi-level optimization to improve the stability of subgraph-level learning tasks.

**Claims And Evidence:**

The author's statement is somewhat convincing in the paper, but the plausibility and reproducibility of the experimental results need to be further examined.

**Essential References Not Discussed:**

I don't think there are any important references that have not been discussed.

**Experimental Designs Or Analyses:**

The experiment is generally correct, see weekness for details.

**Methods And Evaluation Criteria:**

The proposed methodology has research implications for the study of the issue at hand.

**Other Comments Or Suggestions:**

For more information, see Weaknesses.

**Other Strengths And Weaknesses:**

Strengths：
1.This paper is novel and a good target for research.
2.The article is overall easy to read, although there are some logic problems.

Weakness:
1. The writing logic of the paper is somewhat lacking and lacks clear motivation.
2. This paper uses implicit graph neural networks to solve subgraph learning tasks, but lacks sufficient novelty and contribution.
3. Although the authors give an experimental motivation for using bi-level optimization, the experimental results indicate that smaller gamma values may also lead to instability. Therefore, I still doubt the validity of this motivation and whether the author can provide more rigorous theoretical analysis.
4. Some symbols in this paper lack sufficient explanation and clarification.
5. It seems feasible to convert the fixed-point iteration into bi-level optimization, and I would like to know the difference in the final convergence between these two methods.

**Questions For Authors:**

For more information, see Weaknesses.

**Relation To Broader Scientific Literature:**

I finished reading this paper and for the time being I did not find the main contribution of the paper to be relevant to the wider scientific literature.

**Theoretical Claims:**

I think the authors' proof of Theory 3.8 is essentially correct.

---

> ### Author Rebuttal · Authors · 2025-04-01
>
> We appreciate the reviewer’s detailed feedback. Below are our responses addressing each concern:
>
> ---
>
> ## Logic and Motivation Problems
>
> In the revised manuscript, we will include an expanded discussion on why implicit graph neural networks are particularly suited for subgraph learning tasks. We will clarify how capturing long-range dependencies through fixed-point iterations provides a distinct advantage over conventional methods, thereby reinforcing the motivation behind our approach.
>
> ---
>
> ## Novelty and Contribution of the Paper
>
> While our method builds on implicit graph neural networks, our contribution lies in:
> 1. **Novel Extensions:** Developing new extensions to these models specifically for subgraph learning.
> 2. **Label-Aware Integration:** Integrating label-aware subgraph-level information via a novel hybrid graph framework.
> 3. **Bilevel Optimization Formulation:** Formulating the training as a bilevel optimization problem with convergence guarantees.
>
> These aspects, to our knowledge, are the first to be explored in the subgraph context and lead to significant performance improvements over state-of-the-art subgraph neural networks. We will further elaborate on these contributions in the revision.
>
> ---
>
> ## Motivation for Using Bi-level Optimization
>
> The objective of implicit models can be naturally formalized as a bilevel optimization problem. The lower-level problem involves finding the fixed-point embeddings for the current setting, and the upper-level problem focuses on minimizing the classification loss given the fixed-point embedding from the lower level. Bilevel optimization provides a different angle to train implicit models, offering computational advantages. The stability of implicit models is a very interesting topic for further exploration, and to the best of our knowledge, no work has investigated this area so far.
>
> ---
>
> ## Explanation and Clarification of Symbols
>
> We will check and revise the manuscript to ensure that all symbols are clearly defined.
>
> ---
>
> ## Convergence
>
> Both the implicit differentiation (via fixed-point iteration) formulation and the bilevel optimization formulation often lead to equivalent problems, meaning that optimally solving one also solves the other. However, the resulting algorithms differ in various settings. In our case, there are higher practical computational costs for implicit differentiation, which arise due to the following two differences:
>
> - **Iteration Flexibility:** Bilevel optimization permits the use of minimal fixed-point iterations during the early phases of training—empirically, one iteration per gradient step is often sufficient. In contrast, implicit differentiation requires a relatively larger number of fixed-point iterations at each gradient step to maintain accuracy.
>
> - **Backward Gradient Computation:** Implicit differentiation involves computing backward gradients through an additional fixed-point iteration, which adds extra computational overhead compared to bilevel optimization.
>
> ---
>
> We hope these responses address your concerns and clarify the contributions in our work.

---

> > ### Comment · Reviewer_Red8 · 2025-04-05
> >
> > Thanks for the author's thoughtful reply to the question, and considering the opinions of other reviewers, I decided to raise the original score to 3.

---

### Official Review · Reviewer_hQDC · 2025-03-13

**Overall Recommendation:** 3

**Summary:**

This paper combines the information from both subgraphs and nodes to form a hybrid graph to tackle the subgraph-level graph learning problem.  Instead of directly training a GNN on the hybrid graph, it uses implicit GNN combined with a bilevel optimization way to enhance model performance.  Convergency guarantee is provided for this method.  Experiment results show that this method achieved good performance on the evaluated datasets.

**Claims And Evidence:**

The claim that directly train a implicit GNN on the hybrid graph leads to poor performance is underpinned by the experiments.

**Essential References Not Discussed:**

I am not aware of such.

**Experimental Designs Or Analyses:**

The experiment looks valid to me.

**Methods And Evaluation Criteria:**

I am not familiar with subgraph-level graph learning tasks, so I am not sure whether these datasets are good.

**Other Comments Or Suggestions:**

In Fig 3 (b), the (1) under (S2) should be (2), and its children should be (1), (3), (6).

**Other Strengths And Weaknesses:**

The motivation is not clear to me.  I would appreciate it if the author can elucidate why subgraph neural networks are important.

**Questions For Authors:**

What is the connection between max-zero-one label trick and Weisfeiler-Leman test?

**Relation To Broader Scientific Literature:**

This paper is closely related to a broader scope of implicit and unfolded GNNs, where bilevel optimizations are commonly used.

**Theoretical Claims:**

I checked the proof in the main paper.

---

> ### Author Rebuttal · Authors · 2025-04-01
>
> We thank you for your valuable feedback. Our responses are as follows:
>
> ## Motivation
>
> While traditional graph neural networks focus on node-level or entire graph-level representations, many real-world problems require understanding the structure within parts of a graph. Subgraphs often represent meaningful patterns—like communities, motifs, or functional units—that are lost when only considering individual nodes. Moreover, traditional GNNs perform poorly on subgraph-level tasks due to their limited ability to capture localized and complex structural nuances, resulting in weak generalization when applied directly to subgraph classification or related tasks. Therefore, researchers incorporate subgraph information into their models to achieve better generalization. Previous works only consider structural or membership information of subgraphs, which means they primarily leverage the connectivity patterns or the existence of subgraph membership without integrating additional contextual cues. In contrast, our approach augments this by incorporating label-aware subgraph-level information that enriches the representation. By explicitly modeling both the inherent structure and the associated label information, our method not only differentiates between subgraphs with similar structural features but also captures long-range dependencies and complementary interactions among subgraphs. This holistic integration of node-level and subgraph-level cues ultimately leads to more expressive embeddings and significantly improved performance on subgraph-level predictive tasks.
>
> ---
>
> ### Mistake in Figure
>
> We thank the reviewer for pointing out this mistake. We will correct it.
>
> ---
>
> ### Connection between Max-Zero-One Label Trick and Weisfeiler-Leman Test
>
> Max-zero-one labeling enhances subgraph representation learning by augmenting node features based on their membership in subgraphs. It serves as a relaxation of the zero-one trick [1]—which processes each subgraph individually by assigning binary labels. The zero-one trick is capable of producing embeddings with expressiveness equivalent to the $1$-Weisfeiler-Leman ($1$-WL) test, meaning it can distinguish graph structures as well as the $1$-WL test does. Since the max-zero-one trick relaxes these binary assignments by jointly processing multiple subgraphs, it is inherently weaker. Therefore, its discriminative power is limited to that of the $1$-WL test and cannot exceed it.
>
> ---
>
> [1] Zhang, Muhan, et al. "Labeling trick: A theory of using graph neural networks for multi-node representation learning." *Advances in Neural Information Processing Systems* 34 (2021): 9061-9073.
>
> ---
>
> We hope these responses address your concerns and clarify the contributions and design choices in our work.

---

### Official Review · Reviewer_Qr7p · 2025-03-14

**Overall Recommendation:** 3

**Summary:**

The paper introduces the Implicit Subgraph Neural Network (ISNN), an innovative approach designed to enhance subgraph representation learning. ISNN is the first to use implicit neural network models explicitly for subgraphs, addressing limitations in existing methods, particularly concerning capturing long-range dependencies between subgraphs. The authors formulate subgraph representation learning as a bilevel optimization problem, providing theoretical guarantees for convergence and introducing a computationally efficient training algorithm. Experimental results demonstrate that ISNN significantly outperforms existing state-of-the-art approaches across multiple benchmark datasets.

---

(+) The introduction of implicit neural networks for subgraph representation is original and fills a gap in existing subgraph learning methods.

(+) The paper provides solid theoretical backing for the convergence of their proposed bilevel optimization method.

(+) Extensive experiments on multiple datasets clearly show superior performance in terms of Micro-F1 and AUROC scores compared to existing baselines.

(+) The proposed method demonstrates practical runtime efficiency, addressing scalability concerns.

---

(-) Limited sensitivity analysis and ablation studies on certain critical hyperparameters, potentially affecting the interpretability and broader applicability of the approach.

(-) The effectiveness of various proposed methods for constructing the subgraph-level graph is not deeply explored, as shown by their similar performances, even when random edges are introduced.

(-) The reliance on pretraining for constructing subgraph-level graphs could limit applicability in dynamic or real-time settings.

---

## update after rebuttal

Thanks to the authors for the detailed and clear rebuttal. The new experiments on label-aware subgraph construction are helpful and support your design choices well. The gamma sensitivity results are also useful and show reasonable stability.

I understand the current limitations around dynamic graphs, and I agree it’s a valuable direction for future work. Overall, your responses strengthen the paper, and I still lean toward a weak accept (solid contribution with room to grow).

**Claims And Evidence:**

The claims made regarding performance improvements and computational efficiency are well-supported through clear experimental evidence and theoretical analysis. However, the claims regarding the benefit of explicitly constructed subgraph-level information are less convincing, as experiments suggest similar performance even with random graph construction.

**Essential References Not Discussed:**

The paper extensively covers relevant literature. However, it could benefit from discussing more works on dynamic or evolving graph scenarios (if available) since the current formulation and pretraining step may be restrictive in such contexts.

**Experimental Designs Or Analyses:**

The experimental design is robust, evaluating performance on several real-world datasets and comparing against multiple baselines. The analyses are thorough and valid, with clear performance metrics and adequate runtime comparisons.

**Methods And Evaluation Criteria:**

The proposed methods and evaluation criteria are appropriate for the subgraph classification problem. Utilizing standard benchmarks (PPI-BP, HPO-METAB, HPO-NEURO, and EM-USER) aligns well with community standards, providing convincing comparisons against established baselines.

**Other Comments Or Suggestions:**

- Additional discussion on the scalability of the model to large-scale dynamic networks would strengthen the paper.

- Clarification on hyperparameter sensitivity analysis, especially regarding γ, could enhance the practical utility of the paper.

**Other Strengths And Weaknesses:**

(+) Strong motivation clearly identifying practical limitations of existing subgraph models.

(+) Clear visualization and description of methodological innovations.

(+) Practical significance and clear demonstration of superior performance.

(-) Limited investigation of the impacts of graph construction techniques.

(-) Potential issues in scalability and practicality for real-time updates due to reliance on pretraining.

**Questions For Authors:**

1. How sensitive is ISNN to changes in subgraph-level graph construction methods, beyond the initial random comparison?

2. Can you elaborate on the potential applicability or adaptations required for ISNN in dynamic or streaming graph settings?

3. Could you provide further justification or examples where explicit subgraph-level information significantly impacts performance compared to random edge assignment?

**Relation To Broader Scientific Literature:**

The paper appropriately situates its contributions within existing literature on subgraph neural networks and implicit models. It effectively highlights shortcomings in previous models such as SubGNN, GLASS, and SSNP, clearly articulating how ISNN extends beyond these methods to address specific limitations in capturing subgraph-level information and long-range dependencies.

**Theoretical Claims:**

The correctness of the theoretical claims has been checked, particularly regarding the convergence analysis (Theorem 3.8 and Lemma B.1). No immediate issues were found, and the proofs appear sound and complete.

---

> ### Author Rebuttal · Authors · 2025-04-01
>
> We thank you for your valuable feedback. Our responses are as follows:
>
> ---
>
> ## Sensitivity to Subgraph-Level Graph Construction Methods
>
> In our experiments, we previously compared four subgraph-level graph construction methods—**random**, **neighborhood**, **position**, and **structure**. The goal was to show that these classical methods do not significantly improve classification performance, as they achieve similar results to using random subgraph information. This observation echoes conclusions from prior work like GRASS and SSNP.
>
> To further the comparison, we introduce additional label-aware methods:
> - **class-rand:** For each class, we randomly connect *k* pairs of training supernodes.
> - **Star:** For each class, we select a centroid supernode and connect it with all other supernodes within that class.
>
>
> ### Experimental Results on HPO-METAB
>
> | Metric    | ISNN-rand | ISNN-class-rand | ISNN-Star | ISNN  |
> |-----------|----------|--------------|---------|-------|
> | **F1**    | 0.589     | 0.672           | 0.707     | 0.731 |
> | **AUROC** | 0.876     | 0.901           | 0.916     | 0.924 |
>
> The **ISNN-class-rand** method improves performance over ISNN-rand by using label information to ensure that subgraphs within the same class have similar embeddings, making classification easier. The **ISNN-Star** method further boosts performance by selecting a centroid supernode for each class and connecting it to all other supernodes, reinforcing intra-class similarity, though it introduces more edges. Our **ISNN** method connects only the top-*k* most distant pairs of supernodes, balancing label-aware benefits with a sparse graph structure.
>
>
>
> ---
>
>
> ## Additional Hyperparameter Sensitivity Analysis
>
> We further investigate the sensitivity of hyperparameter **gamma** on ISNN.
>
> ### Ablation Results on Gamma for PPI_BP
>
> | Gamma | F1 (10 Runs)    |
> |-------|-----------------|
> | 0.001 | 0.713 ± 0.014   |
> | 0.01  | 0.726 ± 0.022   |
> | 0.05  | 0.670 ± 0.008   |
> | 0.1   | 0.655 ± 0.007   |
> | 0.5   | 0.654 ± 0.022   |
> | 1     | 0.634 ± 0.025   |
>
> ### Ablation Results on Gamma for EM_USER
>
> | Gamma | F1 (10 Runs)    |
> |-------|-----------------|
> | 0.001 | 0.869 ± 0.012   |
> | 0.01  | 0.808 ± 0.027   |
> | 0.05  | 0.853 ± 0.027   |
> | 0.1   | 0.913 ± 0.031   |
> | 0.5   | 0.857 ± 0.018   |
> | 1     | 0.771 ± 0.035   |
>
> These results show that the best performance occurs when gamma is between **0.01 and 0.1**. For example, PPI_BP achieves its best performance at gamma = $0.01$, while EM_USER peaks at gamma = $0.1$. Gamma values outside this range lead to worse performance, highlighting the importance of using a moderate gamma.
>
>
> ---
>
> ## Applicability and Adaptations for Dynamic or Streaming Graph Settings
>
> **ISNN** is designed to capture long-range dependencies via implicit iterations and a hybrid graph framework. To achieve this, the model requires access to the entire graph to compute the fixed-point embedding. Therefore, directly adapting ISNN to dynamic or streaming settings is challenging. However, if a smooth condition on the embeddings can be ensured (for example, bounding the change in the fixed-point embedding under edge perturbations), the error in approximating the final fixed-point embedding could be controlled. This approach offers a potential pathway to adapt ISNN to dynamic settings, which we plan to investigate in future work.
>
> ---
>
> ## Justification for Label-aware Subgraph-Level Information Versus Random Edge Assignment
>
> Our approach, **ISNN**, significantly outperforms random edge assignment. Please compare the performance of ISNN on the HPO-METAB dataset in Table 3 against that of ISNN-rand in Table 4 of our manuscript.
>
> To further illustrate the benefit of our label-aware subgraph information, consider the following example: if two subgraphs with different labels are connected through random edge assignment, the GNN is forced to make their embeddings more similar, which complicates the classification task. Our label-aware design, on the other hand, reinforces the inherent differences between subgraphs, thereby facilitating more discriminative embeddings.
>
> ---
>
> ## Scalability of Our Method
>
> The complexity of a standard GCN is given by:
>
> $$ O(n \cdot d^2 + E \cdot d) $$
>
> where:
> - $n$ is the number of nodes,
> - $E$ is the number of edges, and
> - $d$ is the hidden dimension.
>
> For our method, the complexity becomes:
>
> $$ O((n+s) \cdot d^2 + (E+k) \cdot d) $$
>
> where:
> - $s$ is the number of subgraphs, and
> - $k$ is the number of additional edges introduced.
>
> Thus, the additional overhead introduced by ISNN is an additive term of:
>
> $$ O(s \cdot d^2 + k \cdot d) $$
>
> Empirically, $k$ can be controlled to remain small, and $s$, which is part of the input, is typically small in most practical subgraph learning tasks. Therefore, ISNN exhibits scalability similar to that of a standard GCN in most practical scenarios.
>
> ---
>
> We hope these responses address your concerns and clarify the contributions and design choices in our work.

---

### Decision · Program_Chairs · 2025-05-01

**Decision:**

Accept (poster)

**Comment:**

This paper proposes an implicit neural network model for subgraph representation learning. Model training is formulated as a bilevel optimization problem and an algorithm is provided to find a solution to the optimization problem.

This paper was reviewed by four expert reviewers. The reviewers raised concerns about the scalability of the proposed method, the clarity of the presentation, the method's sensitivity to the graph construction method, and missing baselines and comparisons. Some reviewers also raised concerns about the novelty of the proposed model. After rebuttal, all the reviewers are positive about this paper. Overall, I personally agree with the reviewers that this work is a direct application of well-established techniques to the problem of subgraph representation learning. Although there remains room for improvements, I also agree that the contributions are solid and interesting for the community, and therefore I recommend acceptance. I strongly encourage the authors to revise the paper and improve the writing taking into account the detailed comments in the reviews.